# Diversity and Genetic Characteristics of Astroviruses from Animals in Yunnan Province

**DOI:** 10.3390/v14102234

**Published:** 2022-10-12

**Authors:** Xingyu Huang, Jiuxuan Zhou, Yutong Hou, Rui Wang, Qian Li, Yixuan Wang, Ruiling Yuan, Peng Chen, Binghui Wang, Xueshan Xia

**Affiliations:** 1Faculty of Life Science and Technology, Kunming University of Science and Technology, Kunming 650500, China; 2Yunnan Academy of Forestry and Grassland, Kunming 650201, China

**Keywords:** Astrovirus, genetic diversity, cross-species transmission

## Abstract

Astroviruses (AstVs) are single-stranded RNA viruses, including two main genera: Mamastroviruses (MAstVs) and Avastroviruses (AAstVs). AstVs have been detected in more than 80 different mammals and birds, with the characteristics of multiple cross-species transmission and gene recombination. All these have accelerated the process of virus mutation and posed a potential threat to human beings and animal husbandry. Yunnan province is a global hotspot with rich biodiversity and abundant animal resources and an important area with significance for public health and security because it neighbors a few Southeast Asian countries. This study collected 860 samples from 13 species of animals in Yunnan province for AstVs detection. The results showed that the positive rate of AstVs was 6.05%, and its extremely high genetic diversity was observed in different animal species. Potential cross-species transmission events were also detected from rodents to birds. Therefore, AstVs, which are widely distributed with highly diverse genes and the risk of cross-species transmission to people, deserve more attention in this region.

## 1. Introduction

Astrovirus (AstV) is a non-enveloped single positive-stranded RNA virus with a genome size between 6.4 kb and 7.3 kb [1], synthesized by RNA-dependent RNA polymerase (RdRp), and the genome consists of three open reading frames (ORFs), namely ORF1a, ORF1b, and ORF2 [2]. ORF1 encodes viral nonstructural polyproteins, including RdRp, which may play a key role in viral replication, and ORF2 encodes a capsid protein precursor, which is involved in cell adhesion, mediating cell invasion and stimulating the host’s protective immune response [3]. AstVs are divided into two genera, *Mamastroviruses* (*MAstVs*) and *Avastroviruses* (*AAstVs*) [4]. According to the gene classification method based on ORF2 and developed by the International Association for Taxonomy of Viruses, there are currently 19 kinds of viruses in the genus MAstV, including *MAstV-1*~*MAstV-19*, and 3 kinds of AAstV, including *AAstV-1*~*AAstV-3* [5]. They have different mammalian and bird host sources and can cause varying degrees of enteritis and other diseases through fecal–oral transmission. More than 80 species of birds and mammals are natural hosts of AstVs [6]. At present, AstVs have been successfully isolated from the feces of numerous mammals (including humans, cats, cattle, deer, dogs, mice, pigs, sheep, minks, bats, cheetahs, rabbits, and marine mammals) and birds (including turkeys, chickens, ducks, guinea fowl, and pigeons) [7], but they are not easy to readily grow in laboratory host systems.

Human Astroviruses (HAstVs) mainly cause infantile diarrhea but may also cause diseases in older adults and people with low immune function [8]. The incidence of mammals is mainly mild gastrointestinal symptoms, but there are also reports of encephalitis symptoms [9]. However, AstVs can cause poultry enteritis death syndrome, broiler growth retardation syndrome, kidney and visceral gout in broilers, and fatal hepatitis in ducklings [10]. In the past half of the century, AstVs have caused severe epidemic diseases in many different hosts [11]. These diseases caused by AstVs may seriously affect the growth and development of poult, resulting in stunting, increased mortality, and significant economic losses.

In recent years, new emerging and reemerging zoonotic infectious diseases have occurred frequently, causing heavy losses to human health and property. It has become one of the biggest public health concerns in the world. Wild animals, such as bats, mice, and birds, play essential roles in zoonotic diseases. They are hosts for many zoonotic pathogens. Therefore, carrying out epidemic surveillance of the diseases in wild animals can enhance our understanding of the virus before it becomes seriously pathogenic [12] to prevent its effect in advance and effectively protect humans. Yunnan province is rich in wild animal resources, but there are few studies on AstVs currently. In the context of high disease incidence, it is necessary to study the prevalence and phylogenetic relationship of AstVs in wild animals so as to better understand the genetic diversity and epidemic situation of AstVs and obtain more information on their ecology and evolution. We have conducted a surveillance study of AstVs in some mammals and poultry in Yunnan province. In this study, we identified several new AstVs on the basis of the comparative genome and phylogenetic analysis and found potential cross-species transmission characteristics.

## 2. Materials and Methods

Thirteen species of animals in Yunnan Province were covered in this study, including *Sus scrofa*, bats, birds, etc. The specimens of *Macaca mulatta*, *Sciurus*, *Ursus thibetanus,* and *Vicugna pacos* were from an animal shelter; the specimens of *Cervus nippon* and *Moschus berezovskii* were bred in captivity; the specimen of rodents were caught in the natural environment near towns and villages. Swab samples were collected from protected animals such as *Sus scrofa*, *Macaca mulatta*, etc.; fresh feces samples were collected from birds and bats, and tissue samples were collected from unprotected animals such as rodents. All samples were immersed in a viral transport medium, transported with dry ice, and stored at −80 °C for further laboratory processing.

The suspension was prepared from the tissue, swab, and feces of animals by grinding and centrifugation. Total RNA was extracted from 200 µL of homogenates using a Tianamp virus RNA kit (Tissue, swab) or MagMAX^TM^ virus extraction kit (feces) according to the manufacturer’s instructions. Species of birds and bats were identified by *Cytochrome oxidase I* gene and the *Cytochrome B* gene [13,14]. AstVs were detected by PCR of *RdRp* gene using the method reported previously [15]. The *RdRp* gene was sequenced for further phylogenetic analysis in Tsingke Biological Co. Ltd. Sequences were aligned with the MAFFT. Maximum likelihood analysis was conducted using MEGA-X, with a general time-reversible model (GTR + G + I) and 1000 bootstraps.

## 3. Results and Discussion

In total, 860 samples from 13 species of animals were collected from 13 prefectures or cities of Yunnan province for AstVs detection. Fifty-two samples (6.05%) from six species of animals were positive for AstVs (Table 1). *Sus scrofa* showed the highest infectious rate (41.67%), followed by *Hystrix branchyura* (10.34%), rodents (8.84%), *Larus ridibundus* (8.77%), *Anser indicus* (5.84%), and bats (5.00%). Other seven species were negative for AstVs, and the possible reason is that most of them are captive-bred animals. In other words, *Ursus thibetanus*, *Macaca mulatta*, *Sciurus*, and *Vicugna pacos* were zoo animals; both *Cervus nippon* and *Moschus berezovskii* were bred in captivity through proper introduction channels. The moderate concentration and separation of the breeding environment seemed to prevent them from being infected by AstVs. Previous studies have shown that the detection rates of different species vary considerably [11]. The difference in positive rate was observed among these 13 species, but it was not statistically significant because of the limited number of samples from several rare wildlife. The detection rate of AstVs in different species ranged between 5 and 11%, which was similar to the results in some previous studies [16,17,18,19]. However, there are few reports of *Sus scrofa* infecting AstVs, and the reason for the abnormally high carrier rate in this study may be the limited number of samples. Bats and rodents are the two most studied mammals, and the prevalence and genetic diversity of *MAstVs* should be high. Nevertheless, in this study, higher detection rates of rodents (8.84%) and bats (5.00%) were not shown.

To describe the genetic characteristics of AstVs from wild animals in Yunnan province, all the *RdRp* genes (422bp) were sequenced for phylogenetic analysis. A total of 52 sequences, together with 64 references from the NCBI gene bank, were involved in phylogenetic analysis. As shown in Figure 1a, AstVs are divided into two genera, namely *MAstVs* and *AAstVs*, which indicates that they have been differentiated for a long time. An evolution analysis based on nucleic acid and amino acid sequences showed that MAstVs and AAstVs diverged approximately 310 million years ago [20].

Twenty-seven strains from mammals and three from birds in this study fell into five monophyletic groups of MAstVs genera. Two strains from the bat (CHN/YN/Km-1/2021 and CHN/YN/Cj-4/2021) formed a cluster with the bat AstVs found in other regions of China and Germany, while the strain CHN/YN/Pr-7/2021 showed a very close genetic relationship to the strains found in Singapore. The strains from rodents were clustered into several groups, and our strains were located in two lineages formed by AstVs from *Rattus norvegicus* and wild rodents separately [19]. Five strains of porcine (*Sus scrofa*) relate to porcine AstVs type 2, together with these three strains from porcupine *Hystrix branchyura*, which is a far apart species from porcine. Similar to other RNA viruses, AstVs are thought to be species-specific, but cross-species transmissions are observed frequently [5]. Interestingly, three strains from birds were found to cluster with the MAstVs ones, two strains (CHN/Pzh/435/2021 and CHN/Lyh/22/2021) were closely related to rodent AstVs, and CHN/Pzh/410/2021 formed a cluster with porcine AstVs type 5 (Figure 1b); the amino acid sequence of the RdRp (123aa) form strains CHN/Pzh/435/2021, CHN/Lyh/22/2021, and CHN/Pzh/410/2021 shared high identities of 95.92%, 98.11%, and 99.12% with reference strain (QNJ99366.1, ATP66669.1, and QBJ01354.1). Mammalian *cytochrome B* genes were detected to exclude contamination from other animal samples, which indicates that potential cross-species transmission may occur with other hosts.

Three groups of AAstVs were separated by phylogenetic analysis of these reference sequences from GenBank and the strains detected from wild birds.

In addition to the above 3 virus strains, 22 strains were detected from *Anser indicus* and *Larus ridibundus* in all three groups (Figure 1c). Eleven strains of AAstVs from *Anser indicus* were dispersed in these strains from group 1, including TAstV-1, TAstV-2, DuAstV, DuHV-3, and CAstV, showing multiple sources. The other two strains of *Anser indicus* (CHN/Zt-Dsb/29-2/2021 and CHN/Zt-Dsb/68-2/2021) were phylogenetically related to strains from several wild duck species reported in Hong Kong. Avian Nephritis Virus (ANV) and its closely related lineage share group 2 in the phylogenetic tree. Nine AstVs detected from *Larus ridibundus* were found in this group, closely related to AAstVs detected from *Anas platyrhynchos*, *Anous tenuirostris*, and even *Myotis bechsteinii*.

AstVs are characterized by the genetic diversity of viruses, extensive host sources, and cross-species transmission. However, it has not attracted much attention from virologists since its infection does not cause severe human diseases and large-scale epidemics at present. Little is known about the viral carrier rates and genetic characteristics in animals not related to farming, such as wild animals and animals raised in zoos. Thirteen different wildlife were involved in the detection of AstVs, and the infection rates and genetic characteristics of virus strains are explained in this study. Phylogenetic analysis of 52 strains of viruses from six species of animals revealed the genetic diversity of AstVs and the extensive host origins.

Since the first discovery of AstVs in children, it has been detected in the feces of various mammals and birds. New AstVs sequences have been continuously discovered in various species in recent years [21,22,23,24]. Although the number of AstVs entries with different Taxids in the NCBI Taxonomy database has exceeded 700 (astrovirus-Taxonomy-NCBI (https://www.nih.gov/) URL (accessed on 11 October 2022)), few species are classified by the International Committee for Taxonomy of Viruses (ICTV). There are obvious differences between mammalian virus strains and avian virus strains, but the classification of the two genera is different. Our strains were scattered in the whole phylogenetic tree, and no clustering of the geographical origin or species origin has been observed in this study.

Cross-species transmission is an important way for viruses to break through the inherent host and gain more living space, resulting in greater genetic diversity by subsequent adaptation to new hosts. It is reported that there is a cross-species transmission of AstVs between mammals [20,25,26] and birds [20], but the transmission of MAstVs to birds has not been recorded. The three strains were identified as MAstVs, showing a close genetic relationship with rodent and porcine virus strains, indicating that potential cross-species transmission may also occur between mammals and birds.

The characteristics of genetic diversity, multihost distribution, and cross-species transmission of AstVs make it a potential candidate virus for new emerging zoonotic infections. In addition, further research on its epidemiology, etiology, and cross-species transmission mechanism will help us understand the evolutionary history of the virus kingdom and the driving force behind the genetic diversity of viruses. Yunnan province is the area with the most abundant distribution of animal species in China, and the natural distribution of AstVs must also be prosperous. This study is still limited by the small sample size, and further research is certainly needed to clarify the distribution, genetic characteristics, and transmission chains of AstVs in natural hosts in Yunnan province.

## Figures and Tables

**Figure 1 viruses-14-02234-f001:**
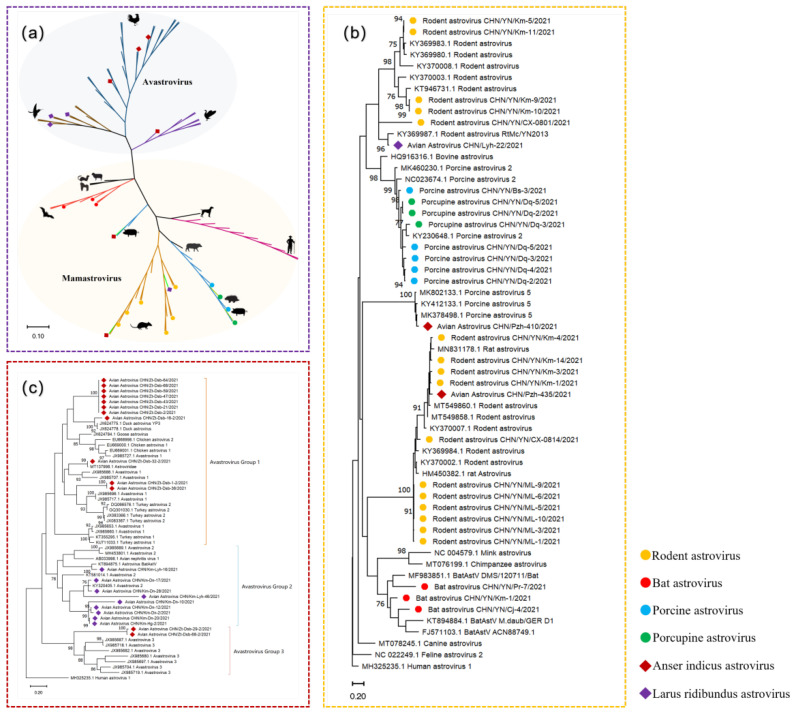
(**a**) According to the partial nucleotide sequences of the *RdRp* gene (369 bp), sequence comparison and clustering were performed with representative AstVs. Maximum likelihood method was used to construct tree (ML), model (GTR + G + I), and bootstrap analysis was performed 1000 times; (**b**) phylogenetic tree based on partial nucleotide sequences of *RdRp* genes (369 bp) of MAstV, model (GTR + G + I); (**c**) phylogenetic tree based on partial nucleotide sequences of *RdRp* genes (369 bp) of AAstV, model (GTR + G + I). The strains obtained in this study are represented by different color symbols.

**Table 1 viruses-14-02234-t001:** Statistical table of samples of mammals and birds in Yunnan province.

Animal Species	No. of Collected Samples	No. of AstVs Positive Samples (%^+^)
**Mammal**		
Bats	60	3 (5.00)
Rodents	181	16 (8.84)
*Sus scrofa*	12	5 (41.67)
*Hystrix branchyura*	29	3 (10.34)
*Ursus thibetanus*	19	0
*Macaca mulatta*	47	0
*Sciurus*	8	0
*Cervus nippon*	20	0
*Moschus berezovskii*	35	0
*Vicugna pacos*	10	0
**Birds**		
*Anser indicus*	257	15 (5.84)
*Larus ridibundus*	114	10 (8.77)
*Grus nigricollis*	68	0

## Data Availability

The data presented in this study are openly available in the NCBI database by accession NO. OP482098—NO. OP482149.

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
