# Peer review of "Diversity and Genetic Characteristics of Astroviruses from Animals in Yunnan Province"

_viruses, 2022, doi:10.3390/v14102234_

Round 1

Reviewer 1 Report

In this study, Huang et al., explore the diversity and frequency of astrovirus infection in animals from 13 different species, including wild as well as captive animals. Of 860 fecal samples analyzed, 52 (6.05%) were found to be positive for astrovirus. They determined the partial nucleotide sequence of the viral RdRp in these samples and generated phylogenetic trees with these 52 sequences together with 64 from reference strains. Although not a strict correlation, a general clustering of viruses according to animal species was observed. Of interest, three avian astrovirus strains grouped with mammalian viruses; two with rodent and one with porcine astroviruses, suggesting cross-transmission between these species. No virus potentially jumping between different mammal species was found. Although cross-species transmission between different mammals has been described, this is the first report of potential cross-species transmission between mammals and birds. With the advent of metagenomic studies astroviruses have been frequently reported in different animal species, and even astrovirus-like viruses have been reported in plants. This type of study is important to understand the genetic diversity, evolution, and potential cross-species transmission of astroviruses.

Minor comments

-Figure 1. Indicate that the color symbols represent the sequences obtained in this study.

-Figure 1. The sequences derived from astrovirus strains detected in Sous scrofa are not identified. Please do.

-Please indicate the region and size of the RdRP that was included in the phylogenetic analyses, and if it represents the full-length or partial sequence of the RdRp gene.

-In figure 1 and in the text they show/mention that 6 species of the 13 evaluated were positive for the virus. However, in line 162, they mention 5, please correct this. Also, one additional species would be Sous scrofa. If this is correct there will be 7 different positive animal species. Please clarify.

Author Response

Dear Reviewer:

Thank you for comments to our manuscript entitled “Diversity and genetic characteristics of astroviruses from animals in Yunnan Province” (ID: viruses-1965429). Those comments are all valuable and very helpful for revising and improving our paper, as well as the important guiding significance to our research. We made a careful revision, and now resubmitting the revised paper to your journal. All revisions in the manuscript are marked in red and located in Track Changes.

Thank you for your consideration on the possible publishing of our article in your journal. We look forward to your kind reply.

Yours sincerely,

Bing-hui Wang,

6 October., 2022

Faculty of Life Science and Technology, Kunming University of Science and Technology,

Kunming, 650500, Yunnan, P.R. China.

Emails: wangbh@kust.edu.cn

The below are the answers point to point:

1.-Figure 1. Indicate that the color symbols represent the sequences obtained in this study.

Response 1: Thank you for pointing this out. We have added a paragraph at the end of Figure 1 legend. (line 61-62)

2-Figure 1. The sequences derived from astrovirus strains detected in Sous scrofa are not identified. Please do.

Response 2: Thank you for this suggestion, but in Figure 1, we use "porcine astrovirus" to represent the strain from Sus scrofa, and I will indicate in the article.

We’ve changed [5 Sus scrofa strains related to porcine AstVs type 2] to [Five strains of porcine (Sus scrofa) related to porcine AstVs type 2] (line 157-158)

3-Please indicate the region and size of the RdRP that was included in the phylogenetic analyses, and if it represents the full-length or partial sequence of the RdRp gene.

Response 3: We agree with the reviewer's assessment. We have indicated the region and size of the RdRp. (line 57, line 60, line 61, line144,).

4-In figure 1 and in the text they show/mention that 6 species of the 13 evaluated were positive for the virus. However, in line 162, they mention 5, please correct this. Also, one additional species would be Sous scrofa. If this is correct there will be 7 different positive animal species. Please clarify.

Response 4: We were really sorry for our careless mistakes. Thank you for your reminder. There are 6 different positive animal species, including Bats, Rodents, Sus scrofa (porcine), Hystrix branchyura (porcupine), Anser indicus and Larus ridibundus. we have corrected the “5” into “6”. (line 193)

Reviewer 2 Report

This article provided some valuable information for understanding the diversity and genetic characteristics of astroviruses, and the cross-species transmission events from rodents to birds deserves attention, but a few modifications are necessary to improve the manuscript.

1.     English language must be revised by a mother tongue English speaker.

2.     Lines 24-25: Cross-species transmission events were also detected from rodents to birds.

Can close genetic relationships indicate that there must be cross species transmission? The author should consider changing cross species transmission to potential cross species transmission.

3.     Lines 38-39: AstVs can be divided into two genera, Mammalian Astrovirus (MAstV) and Avian Astrovirus (AAstV)

All the gene symbols and species/genus names in this paper should change to italic typeface.

4.     Line 41: “three kinds of AAstV, including AVAstv1~AVAstv3”

Please check whether the abbreviation "AVAstv1~AVAstv3" is correct.

5.     Line 91: “RDRP gene was sequenced for further phylogenetic analysis…”and Line33: “which is synthesized by RNA-dependent RNA polymerase (RdRp)”. The abbreviation “RDRP” should be unified.

6.     Line 121: “all the RdRp gene were sequenced for phylogenetic analysis.” change to “all the RdRp genes were sequenced for phylogenetic analysis.”

7.     Line 129: Two strains from the bat (CHIN/YN/Km-1 and CHIN/YN/Jc-4).

The name of the virus strain in the manuscript is inconsistent with that in the figure1b (CHN/YN/Km-1 and CHN/YN/Jc-4), authors should check the full text and modify it.

8.     Line 138:Two strains (CHIN/Pzh/435 and CHIN/Lyh/22) were closely related to rodent AstVs, and CHIN/Pzh/410 formed a cluster with porcine AstVs type 5 (Figure 2b).”

Please show the amino acid homology between avian astrovirus (CHIN/Pzh/435 and 138 CHIN/Lyh/22) and rodent AstVs with close genetic and evolutionary relationship.

9.     Line 163: “AstVs has been detected in the feces of various” change to “AstVs have been detected in the feces of various”.

Author Response

Dear Reviewer:

Thank you for comments to our manuscript entitled “Diversity and genetic characteristics of astroviruses from animals in Yunnan Province” (ID: viruses-1965429). Those comments are all valuable and very helpful for revising and improving our paper, as well as the important guiding significance to our research. We made a careful revision, and now resubmitting the revised paper to your journal. All revisions in the manuscript are marked in red and located in Track Changes.

Thank you for your consideration on the possible publishing of our article in your journal. We look forward to your kind reply.

Yours sincerely,

Bing-hui Wang,

6 October., 2022

Faculty of Life Science and Technology, Kunming University of Science and Technology,

Kunming, 650500, Yunnan, P.R. China.

Emails: wangbh@kust.edu.cn

The below are the answers point to point:

  1. English language must be revised by a mother tongue English speaker.

Response : English language of our manuscripts was revised many times. We did not list the changes here but marked in blue in the revised paper.

  1. Lines 24-25: Cross-species transmission events were also detected from rodents to birds. Can close genetic relationships indicate that there must be cross species transmission? The author should consider changing cross species transmission to potential cross species transmission.

Response : We’ve changed [Cross-species] to [Potential Cross-species]. (line28, line97, line 169 line 215)

  1. Lines 38-39: AstVs can be divided into two genera, Mammalian Astrovirus (MAstV) and Avian Astrovirus (AAstV). All the gene symbols and species/genus names in this paper should change to italic typeface.

Response: They were corrected in our revision paper.

  1. Line 41: “three kinds of AAstV, including AVAstv1~AVAstv3” Please check whether the abbreviation "AVAstv1~AVAstv3" is correct.

Response: It was corrected in our revision paper.(line 47)

  1. Line 91: “RDRP gene was sequenced for further phylogenetic analysis…”and Line33: “which is synthesized by RNA-dependent RNA polymerase (RdRp)”. The abbreviation “RDRP” should be unified.

Response: They were corrected in our revision paper.

  1. Line 121: “all the RdRp gene were sequenced for phylogenetic analysis.” change to “all the RdRp genes were sequenced for phylogenetic analysis.”

Response: As suggested by the reviewer, we have changed [all the RdRp gene were sequenced for phylogenetic analysis] to [all the RdRp genes (422bp) were sequenced for phylogenetic analysis] (line 144).

  1. Line 129: Two strains from the bat (CHIN/YN/Km-1 and CHIN/YN/Jc-4). The name of the virus strain in the manuscript is inconsistent with that in the figure1b (CHN/YN/Km-1 and CHN/YN/Jc-4), authors should check the full text and modify it.

Response: Thanks for your careful examination. We have made the corrections to make the name of the virus strain in the manuscript consistent with that in figure 1.

  1. Line 138: “Two strains (CHIN/Pzh/435 and CHIN/Lyh/22) were closely related to rodent AstVs, and CHIN/Pzh/410 formed a cluster with porcine AstVs type 5 (Figure 2b).” Please show the amino acid homology between avian astrovirus (CHIN/Pzh/435 and 138 CHIN/Lyh/22) and rodent AstVs with close genetic and evolutionary relationship.

Response: A more specific description of amino acid homology between avian astrovirus and rodent AstVs.

The amino acid sequence of the RdRp (123aa) form strains (CHN/Pzh/435/2021, CHN/Lyh/22/2021 and CHN/Pzh/410/2021) shared high identities of 95.92%, 98.11% and 99.12% with reference strain (QNJ99366.1, ATP66669.1 and QBJ01354.1)”. (line164-166)

  1. Line 163: “AstVs has been detected in the feces of various” change to “AstVs have been detected in the feces of various”.

Response: Thank you for your comments. We have changed “AstVs has been detected in the feces of various mammals and birds” to “it has been detected in the feces of various mammals and birds” (line 195).
